# One Step Closer to Enigmatic USCα Methanotrophs: Isolation of a *Methylocapsa*-like Bacterium from a Subarctic Soil

**DOI:** 10.3390/microorganisms11112800

**Published:** 2023-11-17

**Authors:** Olga V. Danilova, Igor Y. Oshkin, Svetlana E. Belova, Kirill K. Miroshnikov, Anastasia A. Ivanova, Svetlana N. Dedysh

**Affiliations:** Winogradsky Institute of Microbiology, Research Center of Biotechnology of the Russian Academy of Sciences Leninsky Ave. 33/2, Moscow 119071, Russia; vinnigo@gmail.com (O.V.D.); ig.owkin@gmail.com (I.Y.O.); svet-bel@mail.ru (S.E.B.); ivanovastasja@gmail.com (A.A.I.)

**Keywords:** atmospheric methane consumption, polar regions, methanotrophic bacteria, *Methylocapsa*, USCα group

## Abstract

The scavenging of atmospheric trace gases has been recognized as one of the lifestyle-defining capabilities of microorganisms in terrestrial polar ecosystems. Several metagenome-assembled genomes of as-yet-uncultivated methanotrophic bacteria, which consume atmospheric CH_4_ in these ecosystems, have been retrieved in cultivation-independent studies. In this study, we isolated and characterized a representative of these methanotrophs, strain D3K7, from a subarctic soil of northern Russia. Strain D3K7 grows on methane and methanol in a wide range of temperatures, between 5 and 30 °C. Weak growth was also observed on acetate. The presence of acetate in the culture medium stimulated growth at low CH_4_ concentrations (~100 p.p.m.v.). The finished genome sequence of strain D3K7 is 4.15 Mb in size and contains about 3700 protein-encoding genes. According to the result of phylogenomic analysis, this bacterium forms a common clade with metagenome-assembled genomes obtained from the active layer of a permafrost thaw gradient in Stordalen Mire, Abisco, Sweden, and the mineral cryosol at Axel Heiberg Island in the Canadian High Arctic. This clade occupies a phylogenetic position in between characterized *Methylocapsa* methanotrophs and representatives of the as-yet-uncultivated upland soil cluster alpha (USCα). As shown by the global distribution analysis, D3K7-like methanotrophs are not restricted to polar habitats but inhabit peatlands and soils of various climatic zones.

## 1. Introduction

Emissions of the potent greenhouse gas methane (CH_4_) from permafrost-affected areas in the Arctic and sub-Arctic regions have received significant research attention due to their potential to increase the atmospheric methane burden as permafrost thaws [1,2,3,4]. As shown by the recent analyses, this methane source might have been overestimated without considering high-affinity (or atmospheric) methane oxidation in soils in high northern latitudes [5,6]. The first report of atmospheric CH_4_ uptake by permafrost-affected Alaskan tundra soils was published in 1990 [7]. Since that time, atmospheric CH_4_ uptake has been reported for both cryosols of high organic carbon and polar desert mineral cryosols [8,9,10,11].

Atmospheric CH_4_ uptake in these ecosystems is due to the activity of aerobic methanotrophic bacteria, which utilize methane as a source of carbon and energy [12,13,14,15,16]. A key enzyme of the methanotrophic metabolism is particulate methane monooxygenase (pMMO), which is present in most currently described methanotroph species. Accordingly, the *pmoA* gene coding for the active-site polypeptide of pMMO is the most frequently used molecular marker in the cultivation-independent detection of aerobic methanotrophs [17].

Attempts to identify atmospheric CH_4_-oxidizers thriving in terrestrial Arctic and sub-Arctic ecosystems have been made in a number of studies that employed a wide array of cultivation-independent molecular approaches [9,11,18,19]. Notably, nearly all of these studies reported the occurrence of “high-affinity” methanotrophic bacteria from an as-yet-uncultivated group called upland soil cluster *Alphaproteobacteria* (USCα) after Knief et al. [20]. The first evidence for the existence of these methanotrophs was obtained for boreal upland soils that demonstrated active atmospheric CH_4_ oxidation [21]. The *pmoA* sequences retrieved in this study could not be affiliated to any of the earlier described methanotrophs and displayed only a distant relationship to *pmoA* gene from *Methylocapsa acidiphila* B2, an alphaproteobacterium isolated from acidic peat [22,23]. The *pmoA* sequences of USCα methanotrophs have been recovered from many acidic and near-neutral boreal, subarctic and arctic upland soils [9,19,20,24,25,26].

Since methanotroph affiliation with the USCα group has long been defined by the *pmoA* phylogeny, the exact taxonomic position of these bacteria remained unknown for a long time and no 16S rRNA gene sequences were available for members of this group. The reconstruction of the first draft genome of a USCα methanotroph from a boreal acidic soil revealed its affiliation to the alphaproteobacterial family *Beijerinckiaceae* and its close phylogenetic relationship with methanotrophic bacteria of the genus *Methylocapsa* [27]. The metagenomic assembly obtained in this study was classified as belonging to a new taxon, which was tentatively named *Candidatus* Methyloaffinis lahnbergensis.

Further isolation of a pure culture of the first cultivated member of the USCα clade, strain MG08, provided valuable insights into the physiology and metabolism of these methanotrophs [28]. Strain MG08 showed the ability to grow at atmospheric concentrations of CH_4_ and to assimilate carbon from both CH_4_ and CO_2_. The ability to fix N_2_ and to oxidize other atmospheric trace gases, CO and H_2_, were also confirmed for this methanotroph [28]. Apparently, these bacteria thrive in arctic habitats due to the ability to scavenge atmospheric trace gases. Based on the results of comparative genome analysis, strain MG08 was classified as a member of the genus *Methylocapsa* and was tentatively named ‘*Methylocapsa gorgona*’ MG08. Although the *pmoA* sequence of ‘*Methylocapsa gorgona*’ MG08 clusters within the radiation of USCα *pmoA* clade, it does not correspond to the major group of *pmoA* sequences, which are most often retrieved from upland soils.

Several metagenome-assembled genome sequences (MAGs) of USCα-related methanotrophs have been retrieved from high Arctic ecosystems. One of them, Canadian MAG AHI, was obtained from an acidic carbon-poor cryosol, which was shown to consistently consume atmospheric methane at Axel Heiberg Island in the Canadian high Arctic [11]. The other two MAGs, named USC1 and USC2, were assembled from the active layer of a permafrost thaw gradient in Stordalen Mire, Abisko, Sweden [18]. Although these MAGs were addressed as ‘closely related to USCα’ group, they did not fall within the radiation of ‘true’ USCα methanotrophs represented by *Candidatus* Methyloaffinis lahnbergensis.

In this study, we report isolation and characterization of a *Methylocapsa*-like bacterium, strain D3K7, which represents this earlier uncultivated subgroup of USCα-related methanotrophs from Arctic and sub-Arctic terrestrial habitats. Strain D3K7 was isolated from a sub-Arctic sandy loam soil of northern Russia and formed a common phylogenetic clade with the above discussed MAGs retrieved from the Canadian and Swedish high Arctic. This clade occupies a phylogenetic position in between characterized *Methylocapsa* methanotrophs (including ‘*Methylocapsa gorgona*’ MG08) and as-yet-uncultivated ‘true’ representatives of the USCα group.

## 2. Materials and Methods

### 2.1. Sampling Site

The samples of sandy loam soil used in this study were collected in July 2019 from a subarctic region of Northern Russia. The sampling site was located on the shore of the River Kosaya, a right tributary of the River Dudinka, Northwestern Russia, Krasnoyarsk Krai (69.4162 N, 86.5324 E) (Figure 1a). The sparse vegetation cover was composed of *Equisétum*, *Lycopódium*, *Chamaenerion*, and *Polytrichum* species; *Salix arctica*, *Bétula nána*, *Pinus* sp. and *Álnus* sp. were also present (Figure 1b). No distinctive surface organic layer was seen in the soil profile (Figure 1c). Three individual soil samples (~500 g each) were collected from three sampling plots located at a distance of 40–50 m from each other. The samples were transported to the laboratory and used for further molecular analyses and cultivation experiments.

### 2.2. Soil DNA Extraction and High-Throughput Sequencing of 16S rRNA Genes

Total DNA was isolated from 0.25 g of soil samples using FastDNA SPIN Kit for Soil (MP Biomedicals, Heidelberg, Germany) according to the manufacturer’s instructions. The V3-V4 variable region of the prokaryotic 16S rRNA genes was obtained by PCR with primers 341F (5′-CCTAYGGGDBGCWSCAG) and 806R (5′-GGACTACNVGGGTHTCTAAT) [29]. PCR fragments were barcoded using Nextera XT Index Kit v2 (Illumina, San Diego, CA, USA). The PCR fragments were purified using Agencourt AMPure Beads (Beckman Coulter, Brea, CA, USA) and quantified using Qubit dsDNA HS Assay Kit (Invitrogen, Carlsbad, CA, USA). Then all the amplicons were pooled together in equal molar amounts and sequenced on the Illumina MiSeq instrument (2 × 300 nt reads). The pool of 16S rRNA gene sequences was analyzed with QIIME2 v.2019.10 (https://qiime2.org, accessed on 20 June 2023) [30]. Q-score and VSEARCH uchime plugin were used for sequence quality control, denoising and chimera filtering [31,32]. Operational Taxonomic Units (OTUs) were clustered by applying the VSEARCH plugin [32] with open-reference function using SILVA 138 SSU database [33] with 97% identity and following singleton removal. Taxonomy assignment was performed using BLAST against Silva v. 138 database with 80% identity.

### 2.3. Enrichment and Isolation of Methane-Oxidizing Bacteria

The liquid mineral medium M2 (pH 6.0–6.3), which was used in our previous studies for isolation of *Methylocapsa* species from northern terrestrial habitats [23,34], was employed for obtaining methane-oxidizing microbial consortia from the studied subarctic soil. In contrast to our previous studies, this medium was supplied with 10% (*v*/*v*) of the soil extract. The latter was prepared by mixing 100 g of soil with 500 mL of distilled water, shaking the resulting soil suspension for 1 h at 120 rpm, centrifuging it at 8000 rpm for 10 min, and using sediment-free water phase as a medium supplement. A set of three 500-mL serum bottles filled with 50 mL of the mineral medium with soil extract (gas phase: liquid phase ratio of 9:1) were used for these incubation experiments. We used 2 g of soil to inoculate each of these bottles, after which they were sealed with butyl rubber septa. Methane and CO_2_ were injected in the bottle headspaces using syringes equipped with disposable filters (0.22 µm) up to final concentrations of 10 and 1% (*v*/*v*), respectively, and the bottles were incubated horizontally at 20 °C without shaking in the dark. The gas phase in the bottles was replaced with a freshly prepared gas mixture once in 2 months. After six months of incubation, aliquots of the culture suspensions from the bottles were sampled for microscopic examination. One bottle containing a well-developed microbial consortium dominated with small, slightly curved cells of *Methylocapsa*-like morphology was selected for preparing a dilution series. Aliquots of this dilution series were streaked on polycarbonate filters (Nucleopore Track-Etch membrane with pore size 0.2 µm). Inoculated filters were placed in the wells of the 6-well plate filled with the liquid medium M2 and were incubated at 20 °C in desiccators containing approximately 30% (*v*/*v*) methane and 5% CO_2_ (*v*/*v*) in the air. A similar approach was used before for isolating a pure culture of ‘*Methylocapsa gorgona*’ MG08 [28]. The colonies appearing on the filters were picked and restreaked on the new filters until colonies containing morphologically uniform cells were obtained. One of these colonies was transferred back to the serum bottle with the liquid medium M2 and designated strain D3K7. The culture purity of this isolate was verified by examination using phase-contrast microscopy and by plating on 10-fold diluted Luria–Bertani agar (1.0% tryptone, 0.5% yeast extract, 1.0% NaCl).

### 2.4. Microscopic Studies

Morphological observations and cell-size measurements were made with a Zeiss Axioplan 2 microscope and Axiovision 4.2 software (Zeiss, Oberkochen, Germany). The electron microscopy analysis was performed as described in our recent study [35].

### 2.5. Growth Experiments

Physiological tests were carried out on cultures grown in liquid medium M2 with methane as the growth substrate. The growth of strain D3K7 was monitored by nephelometry at 600 nm using a “BioPhotometer” spectrophotometer (Eppendorf) for 2 weeks under a variety of conditions, including temperatures of 2–37 °C, pH values of 3.8–8.5, and NaCl concentrations of 0–1.0% (*w*/*v*). The following carbon sources (each at a concentration of 0.05% *w*/*v*) were examined to determine the range of substrates utilized by strain D3K7: methanol, ethanol, formate, formaldehyde, glucose, fructose, arabinose, lactose, sucrose, maltose, galactose, acetate, citrate, oxalate, malate, pyruvate, acetate and succinate. The capacity to utilize methanol at concentrations from 0.01 to 1% (*v*/*v*) was determined in liquid M2 medium supplemented with methanol.

Growth dynamics of strain D3K7 in the presence of two substrates, methane and acetate, were examined using 120 mL screw-cap serum bottles filled with 10 mL of M2 medium (pH 6.5) with/or without 0.05% (*w*/*v*) of sodium acetate. Methane (where necessary) was injected in the bottle headspaces using syringes equipped with disposable filters (0.22 µm) up to final concentrations of 10% (*v*/*v*). Bottles were inoculated with cells of strain D3K7 and incubated on a rotary shaker (120 r.p.m.) at 22–25 °C for 5 days.

The ability of strain D3K7 to oxidize methane when the latter is present in low concentrations (below 100 p.p.m.v) was assessed using two types of experimental incubations: (1) conventional liquid cultures of strain D3K7 and (2) bottles with adsorbent materials (perlite or vermiculite) inoculated with cells of strain D3K7. In the first case, 120 mL flasks were filled with 4 mL of grown culture of strain D3K7 (OD_600_ 0.8) and 1 mL of fresh M2 medium (pH 6.5) with or without 0.05% (*w*/*v*) sodium acetate. Methane (where necessary) was injected in the flasks up to the concentrations of ~100 p.p.m.v. Flasks were incubated under static conditions at 22 °C for 3 weeks.

In the second case, 3 mL of cell suspensions (see above) were added to 120 mL screw-cap flasks containing 300 mg sterilized perlite or vermiculite granules. Methane was injected in the headspace up to the final concentration of ~100 p.p.m.v. Control incubations were prepared in the same way but were not inoculated with cells of strain D3K7. The flasks were incubated at 22 °C for 3 weeks. Gas samples were taken from experimental flasks at regular time intervals for determination of CH_4_ concentrations using a Crystal 5000.1 gas chromatograph (Chromatec, Joshkar-Ola, Russia).

### 2.6. Genome Sequencing and Annotation

The culture of strain D3K7 was grown in the liquid M2 as described above. The cells were harvested after incubation at 20 °C for 14 days. Genomic DNA extraction was completed using the standard acetyltrimethyl ammonium bromide (CTAB) and phenol-chloroform protocol [36]. The genomic paired-end (2 × 300) library was prepared with a NEBNext ultra II DNA Library kit (New England Biolabs) and sequenced using a MiSeq instrument (Illumina, San Diego, CA, USA). Primer sequences were removed from the Illumina reads using Cutadapt v.1.17 [37] with the default settings, and low quality read ends were trimmed using Sickle v.1.33 (option q = 30) (https://github.com/najoshi/sickle accessed on 18 July 2023). Nanopore sequencing library was prepared using the 1D ligation sequencing kit (SQK-LSK108, Oxford Nanopore, UK). Sequencing was performed on an R9.4 flow cell (FLO-MIN106) using MinION device. The hybrid assembly of short and long reads was performed using Unicycler v.0.4.8 [38] and refined using Pilon v1.24 [39]. Assemblies were evaluated with Quast 5.0 [39] and Busco 5.1.2 [40]. Annotation was performed using the Prokaryotic Genome Annotation Pipeline [41].

### 2.7. Phylogenomic Analysis and Genome-Encoded Features

The genome-based phylogeny of strain D3K7 was determined based on the comparative analysis of 120 ubiquitous single-copy proteins using the Genome Taxonomy Database [42] and GTDB-Toolkit v. 2.0. [43]. The maximum likelihood genome-based tree was constructed using MegaX software [44]. Genes of key metabolic pathways were manually verified using blastp searches [45]. Pangenomic comparison of currently described *Methylocapsa* species and *Methylocapsa*-related MAGs was performed using the Roary package [46]. The pool of examined genomes included those from strain D3K7, *Candidatus* Methyloaffinis lahnbergensis USCa_MF, ‘*Methylocapsa gorogona*’ MG08, *Methylocapsa aurea* KYG^T^, *Methylocapsa acidiphila* B2^T^, *Methylocapsa palsarum* NE2^T^, and the Canadian MAG AHI, which was phylogenetically most closely related to our new isolate. As a criterion for gene orthology, a threshold of 50% amino acid sequence identity was used.

### 2.8. Global Distribution Analysis

In order to assess the occurrence of strain D3K7-like bacteria in various environments, the Integrated Microbial Genomes (IMG/M) database of the Joint Genome Institute and the GeneBank database were screened for *pmoA* sequences similar to that in strain D3K7 (sequence similarity ≥ 85%, 250 bp subject length match). The full-length 16S rRNA gene sequence from strain D3K7 was used to screen Short Read Archive (SRA) datasets with IMNGS platform [47] using a minimum threshold of 99% identity and a minimum size of 200 nucleotides. The longitude, latitude and habitat descriptions of resulted hits were extracted, and locations mapped in R with mollweide projection (packages: maps, mapdata, maptools, mapproj).

### 2.9. Sequence Accession Numbers

The raw data generated from 16S rRNA gene sequencing were deposited in Sequence Read Archive (SRA) under the accession number BioProject PRJNA1026465. The genome sequence of strain D3K7 has been deposited in NCBI GenBank under the accession number CP123229.1.

## 3. Results

### 3.1. Prokaryote Diversity and Methanotrophic Bacteria Identified in a Subarctic Soil

A total of 166,098 partial 16S rRNA gene sequences (mean amplicon length 253 bp) were retrieved from the examined subarctic tundra soil. Of these, 124,821 reads were retained after quality filtering, denoising, and removing chimeras. The relative abundance of *Archaea*-affiliated 16S rRNA gene fragments was low (≤0.3% of total reads). Predominant bacterial phyla identified in the examined soil samples were the *Acidobacteriota* (25.8 ± 1.2% of the total number of 16S rRNA gene fragments, mean ± SE), *Proteobacteria* (21.7 ± 2.1%), *Verrucomicrobiota* (12.5 ± 1.3%), *Actinobacteriota* (10.2 ± 3.0%), *Chloroflexota* (7.6 ± 1.2%), *Bacteroidota* (6.5 ± 1.2%), and *Gemmatimonadota* (4.5 ± 0.8%) (Figure 2).

Minor bacterial groups (with relative abundance below 3%) included *Patescibacteria*, *Planctomycetota*, *Myxococcota*, *Elusimicrobiota*, *Cyanobacteriota*, and *Bdellovibrionota*. The pool of *Proteobacteria*-affiliated 16S rRNA gene fragments was examined for the presence of sequences belonging to gammaproteobacterial (order *Methylococcales*) and alphaproteobacterial (order *Rhizobiales*) methanotrophs. No *Methylococcales*-affiliated reads were retrieved from the examined soil. By contrast, *Rhizobiales*-affiliated 16S rRNA gene fragments comprised nearly half of all proteobacterial reads (Figure 2). The second largest family within the *Rhizobiales* was *Beijerinckiaceae* (21.3 ± 11.9% of all *Rhizobiales*-related sequences). The latter was represented by methanotrophs of the genera *Methylocystis* and *Methylocapsa*, methylotrophs of the genus *Methylorosula*, and phototrophs of the genus *Roseiarcus*. The most abundant OTU among *Beijerinckiaceae* was classified as representing moss-associated methanotroph *Methylocystis bryophila* (67% of all *Beijerinckiaceae*-affiliated reads) [48]. *Methylocapsa*-related reads comprised 4.8% of all *Beijerinckiaceae*-affiliated sequences.

### 3.2. Isolation and Identification of Strain D3K7

After 6 months of incubation under static conditions, the bottles with methane-oxidizing enrichment cultures inoculated with studied soil samples were subjected to microscopic examination. Microbial consortia that developed in all bottles were dominated by *Methylocystis*- and *Methylocapsa*-like cells, which were rod-like to reniform in shape and formed large cell aggregates. These cells varied in size from very small (0.8–1 µm) to those reported for most described *Methylocystis* and *Methylocapsa* species (1.5–3 µm). One enrichment culture, which was dominated with very small cells of *Methylocapsa*-like morphology, was selected for further isolation efforts. Plating cell suspensions onto the surface of polycarbonate filters resulted in formation of several micro-colonies after 2 months of incubation with methane (Figure 3a).

These colonies were composed of short thick rods or coccoids of irregular shape, which were assembled in large cell clusters containing up to several hundred cells (Figure 3b). One of these micro-colonies was picked and used to inoculate 60 mL vial containing 10 mL of the liquid medium M2 and 10% methane in the headspace. The resulting culture was serially diluted in tenfold steps until the target bacterium was obtained in a pure culture and designated strain D3K7. Cells of strain D3K7 were Gram-negative, non-motile, encapsulated short rods or coccoids, 1.4 ± 0.04 μm in length and 0.9 ± 0.02 μm in width. Analysis of ultrathin cell sections revealed the presence of intracytoplasmic membranes located on one side of a cell (Figure 3c) as characteristic for methanotrophs of the genus *Methylocapsa*.

Comparative analysis of 16S rRNA gene sequence of strain D3K7 confirmed affiliation of this isolate with the genus *Methylocapsa*. The closest taxonomically characterized phylogenetic relatives of strain D3K7 were *M. aurea* KYG^T^ and *M. palsarum* NE2^T^ (98.6 and 98.4% 16S rRNA gene sequence similarity, respectively). Strain D3K7 displayed 96.8% 16S rRNA gene sequence similarity to that of ‘*Methylocapsa gorgona*’ MG08 and 96.1% similarity to that of as-yet-uncultivated *Candidatus* Methyloaffinis lahnbergensis.

Since methanotroph affiliation with USCα group has long been defined by the *pmoA* phylogeny, the corresponding comparative analysis of the *pmoA* sequence from strain D3K7 was performed as well (Figure 4).

The latter revealed 69–71% identity to PmoA sequences from described *Methylocapsa* representatives with validly published names, i.e., *M. acidiphila*, *M. palsarum*, and *M. aurea*, while the sequence identity to PmoA from ‘*Methylocapsa gorgona*’ MG08 was 80%. The highest sequence identity of PmoA from strain D3K7, however, was observed with the corresponding protein sequences encoded in metagenomes obtained from subarctic and Antarctic ecosystems (up to 98%). The sequence identity with PmoA from *Candidatus* Methyloaffinis lahnbergensis was 79%.

### 3.3. Growth Characteristics

Like other members of the genus *Methylocapsa*, strain D3K7 grew on methane as the sole carbon and energy source. Growth occurred in a relatively narrow pH range of 5.5–7.8 with the optimum at pH 6.6–7.0. The temperature range for growth was 5–30 °C with the optimum at 20–25 °C. The specific growth rate of strain D3K7 in a liquid culture under methane (10%, *v*/*v*) at 20–22 °C was 0.015 h^−1^ (equivalent to a doubling time of 46.2 h). Maximal OD_600_ of methane-grown cultures of strain D3K7, which was reached after 8–10 days of cultivation, was about 0.9. These growth characteristics are somewhat lower than those reported for earlier described *Methylocapsa* species [23,34,49]. Very poor, nearly undetectable growth was observed on methanol, which supported growth only when used at concentrations below 0.2% (*v*/*v*). Similar to *Methylocapsa acidiphila* B2^T^, strain D3K7 was able to fix dinitrogen and grew in nitrogen-free medium under fully aerobic conditions.

In addition to methane and methanol, strain D3K7 was capable of slow growth (specific growth rate 0.005 h^−1^) on acetate (Figure 5a). Maximal OD_600_ of acetate-grown cultures of strain D3K7 was around 0.2. In the presence of two substrates, methane and acetate, the specific growth rate of this methanotroph increased to 0.017 h^−1^.

No growth was observed on other tested carbon substrates (see Methods, Section 2.5). Strain D3K7 was sensitive to NaCl: 0.3% (*w*/*v*) NaCl inhibited growth by 90% and 0.5% (*w*/*v*). NaCl inhibited growth completely.

### 3.4. Oxidation of CH_4_ Provided in Low Concentrations

Strain D3K7 was examined for the ability to oxidize methane when the latter is present in low concentrations (below 100 p.p.m.v.). Two types of experimental incubations were used: (1) conventional liquid cultures of strain D3K7, and (2) bottles with adsorbent materials (perlite or vermiculite), which were inoculated with cell suspensions of strain D3K7. As seen in Figure 5b, CH_4_ concentration in experimental flasks declined faster if acetate was supplied to strain D3K7 in addition to methane.

A similar effect was observed for experimental bottles containing perlite or vermiculite, which were used to simulate mineral particles in a well-aerated soil. CH_4_ consumption was faster when acetate was supplied as a second available substrate (Figure 6). No decline of CH_4_ concentration occurred in control bottles which were not inoculated with cell suspensions of strain D3K7.

### 3.5. Genome Sequencing and Phylogenomic Placement

The final hybrid genome assembly for strain D3K7 consisted of one circular chromosome with a total length of 4,607,683 bp. No plasmids were detected. The genome contained one *rrn* operon copy (16S-23S-5S rRNA), 49 tRNA genes, and 3701 predicted protein-coding sequences. DNA G + C content in strain D3K7 was 58.0%, which was close to that in ‘*Methylocapsa gorgona*’ MG08 (58.9%) and lower than that in described *Methylocapsa* species with validly published names (61.3–61.8%).

The genome-based phylogeny of strain D3K7 was inferred using the comparative sequence analysis of 120 ubiquitous single-copy proteins (Figure 7).

According to the results of this analysis, our isolate formed a common lineage (bootstrap support of 100%) with metagenome-assembled genomes obtained from the active layer of a permafrost thaw gradient in Stordalen Mire, Abisco, Sweden [18] and the mineral cryosol at Axel Heiberg Island in the Canadian high Arctic [11,50]. This lineage belongs to a broad phylogenetic clade defined by *Candidatus* Methyloaffinis lahnbergensis, a ‘true’ USCα methanotroph isolated from an acidic forest soil in Germany [27] and an USCα-like methanotroph, ‘*Methylocapsa gorgona*’ MG08 [28]. Other two sequences in this clade were represented by the MAG from an Antarctic desert surface soil [51] and the MAG obtained from termitarium soil in Australia (GenBank Bioproject: PRJNA663662). Notably, strain D3K7 occupied a phylogenetic position in between *Candidatus* Methyloaffinis lahnbergensis and all currently characterized *Methylocapsa* methanotrophs, including ‘*M. gorgona*’ MG08.

### 3.6. Pangenome Analysis, Shared and Unique Functional Characteristics

The genomes from strain D3K7, *Candidatus* Methyloaffinis lahnbergensis USCa_MF, ‘*Methylocapsa gorogona*’ MG08, *Methylocapsa aurea* KYG^T^, *Methylocapsa acidiphila* B2^T^, *Methylocapsa palsarum* NE2^T^, and the Canadian MAG AHI, were used for the pangenome analysis. The Roary-based approach clustered protein-coding sequences into core, shell, and cloud genomes. Of 10,779 gene clusters, *Methylocapsa*-like methanotrophs pan-genome core comprised 1097 genes (10.2% of total gene clusters), with the accessory genome containing 2759 gene clusters (25.6% of total gene clusters) in the shell and 6923 in the cloud (64.2% of total gene clusters) (Figure 8).

All examined genomes contained a single *pmoCAB* operon and from one to three orphan *pmoC* genes. Soluble MMO was not encoded in the genomes of these methanotrophs. With the only exception of MAG USCα AHI, all examined genomes of *Methylocapsa*-like methanotrophs contained a single *mxa* operon encoding the classical methanol dehydrogenase and one-two copies of *xox* gene clusters encoding the alternative lanthanide-dependent methanol dehydrogenase. The absence of methanol dehydrogenase-encoding genes in MAG USCα AHI, most likely, is explained by its incompleteness. The *hpnD*, *hpnC* and *hpnE* genes necessary for the synthesis of hopanoids were found in all examined genomes. Also, the fatty acid desaturase and sterol desaturase-encoding genes were found in all genomes of *Methylocapsa*-like methanotrophs. As demonstrated in a recent study [52], the cold-tolerant methanotroph *Methylovulum psychrotolerans* increases synthesis of unsaturated fatty acids and hopanoids with a decrease in temperature. These results demonstrate that fatty acid and hopanoid composition can be remodeled to maintain bacterial membrane homeostasis upon cold adaptation. With the only exception of *Candidatus* Methyloaffinis lahnbergensis USCa_MF, the genomes of all studied methanotrophs contained a complete set of *fli*-genes necessary for the synthesis of flagella. Clusters of *nif* genes encoding the Mo-Fe-nitrogenase complex were found in the genomes of all studied methanotrophs except *Candidatus* Methyloaffinis lahnbergensis USCa_MF and MAG USCa AHI.

The genome of strain D3K7 was also inspected for the presence of genes that allow scavenging of trace gases, such as H_2_ and CO. However, no high-affinity hydrogenases were revealed in strain D3K7. The only hydrogenase identified was the hydrogenase-4, which was shown to be responsible for H_2_ production at low pH upon formate supplementation [53]. Similarly, no gene candidates for carbon monoxide dehydrogenases were found in the genome of strain D3K7. The Calvin–Benson–Bassham cycle was incomplete since no genes encoding forms I, II, and III of RuBisCo, which catalyzes the incorporation of inorganic carbon into 3-phosphoglyceric acid [54], was revealed in the genome of strain D3K7. Instead, the genome contained genes for the RuBisCo-like protein (form IV RuBisCo) that does not catalyze CO_2_ fixation [55]. This protein was demonstrated to use 5-methylthioribulose-1-phosphate as a substrate, resulting in the formation of two products, i.e., 1-thiomethyl-D-xylulose-5-phosphate and 1-thiomethyl-D-ribulose-5-phosphate [56]. The reaction catalyzed by this RLP was suggested to be involved in a new pathway for sulfur salvage. Notably, the genomes of other earlier described *Methylocapsa* species, i.e., *M. acidiphila*, *M. aurea*, and *M. palsarum*, encode form Ic RuBisCo as well as form IV RuBisCo. The presence of solely form IV RuBisCo is a distinctive characteristic found only in strain D3K7 and ‘*Methylocapsa gorgona*’ MG08.

Similar to ‘*Methylocapsa gorgona*’ MG08 [28], strain D3K7 harbors the complete set of genes for reductive glycine pathway (rGlyP), enabling the conversion of CO_2_ and formate into glycine. Thus, genome-encoded traits related to metabolism of trace gases in strain D3K7 were limited to methane oxidation via pMMO and CO_2_ fixation via rGlyP.

### 3.7. Biogeography of Strain D3K7-like Methanotrophs

IMG database searches for *pmoA* gene sequences related to that of strain D3K7 yielded 1040 matches with >85% identity, which corresponds to 42 locations (Figure 9). These sequences originated from a broad range of terrestrial habitats, including various types of peats and soils (forest, grassland, floodplain, bulk). GenBank database added eight hits for the same search parameters set for *pmoA* genes. These matches are also affiliated with wetland and glacier soils (Figure 7). SRA database searches for 16S rRNA genes related to strain D3K7 generated 561 hits with >99% identity threshold. These correspond to 95 locations (Figure 9).

## 4. Discussion

According to the *pmoA*-based phylogeny, and to the results of comparative genome analysis, our new isolate from a subarctic soil, strain D3K7, occupies an intermediate position between all so far described representatives of the genus *Methylocapsa* and members of the USCα group, which elude all cultivation efforts. Isolation of this bacterium, therefore, takes us one step closer to the enigmatic methanotroph group. Notably, strain D3K7 forms a common clade with metagenome-assembled genomes obtained from the active layer of a permafrost thaw gradient in Stordalen Mire, Abisco, Sweden [18], and the mineral cryosol at Axel Heiberg Island in the Canadian high Arctic [11], which were earlier addressed as belonging to USCα methanotrophs. As we see now, this is not fully correct. This phylogenetic clade should rather be addressed as a group of USCα-related methanotrophs that appear to be common in polar terrestrial ecosystems. Our biogeography analysis, however, suggests that these bacteria are not restricted to polar habitats but inhabit peatlands and soils of various climatic zones (see Figure 9).

Apparently, patience was the main secret behind our success in isolating this novel methanotroph. The enrichment procedure continued for six months, the gas phase to liquid phase ratio in cultivation bottles was extended to 9:1, and the bottles were incubated horizontally, under static conditions. The fact that soil extract was added in the cultivation medium, most likely, did not play a role since the isolate did not require any specific growth factors and was able to grow in a common mineral medium used for *Methylocapsa*-like methanotrophs. Cells of strain D3K7 were smaller, and the specific growth rate was lower than those in earlier described *Methylocapsa* species. Yet, this novel isolate displayed many *Methylocapsa*-specific traits, such as psychrotolerance, sensitivity to NaCl, localization of ICM on one side of a cell only, absence of soluble MMO, ability to fix dinitrogen, and to develop at low CH_4_ concentrations. The ability to utilize acetate in the presence and in the absence of methane is one of the key adaptations of strain D3K7 for survival in a habitat where only trace amounts of CH_4_ are available. Notably, the ability to utilize acetate varies among described *Methylocapsa* species. Thus, *M. acidiphila* B2^T^, *M. palsarum* NE2^T^ and ‘*M. gorgona*’ MG08 do not grow on acetate, while *M. aurea* KYG^T^ and strain D3K7 possess this ability. The latter was experimentally confirmed for environmental populations of USCα methanotrophs in an acidic forest soil by using stable-isotope probing of RNA and DNA to investigate the assimilation ^13^C-acetate [57]. That study revealed the incorporation of ^13^C-acetate into USCα *pmoA* mRNA, suggesting that the contribution of alternative carbon sources, such as acetate, to the metabolism of the atmospheric methane oxidizers might be substantial [57]. Additional energy required for growth may be supplied from acetate utilization. As shown in our incubation experiments, oxidation of CH_4_ provided in low concentrations (below 100 p.p.m.v.) by strain D3K7 was more intense in the presence of acetate.

The fact that strain D3K7 is not capable of fixing CO_2_ via the Calvin–Benson–Bassham cycle was somewhat surprising given that all cultivated representatives of the genus *Methylocapsa* (with the only exception of ‘*Methylocapsa gorgona*’ MG08) possess form Ic RuBisCo. The functional role of RuBisCo-like enzyme (or form IV RuBisCo) in ‘*Methylocapsa gorgona*’ MG08 and strain D3K7, therefore, remains to be elucidated. The reductive glycine pathway appears to be the only metabolic route for CO_2_ fixation by these methanotrophs.

Although the reason behind the “nonculturability” of USCα methanotrophs remains unclear, some hints for obtaining these bacteria in cultures are now available. Apparently, these methanotrophs can be enriched under static incubation conditions only. As follows from the study of Tveit and co-authors [28], as well as this work, the use of membrane filters for obtaining target isolates from enrichment cultures is also the only so far reported successful approach. This suggests that cells of USCα methanotrophs should be exposed to air in order to develop in a normal way. The use of various adsorbent materials to simulate mineral particles in a well-aerated soil may also be considered as a potentially suitable approach. The latter, however, needs to be verified in a specialized, long-term study, given the specific nature of these slow-growing atmospheric methane oxidizers.

## Figures and Tables

**Figure 1 microorganisms-11-02800-f001:**
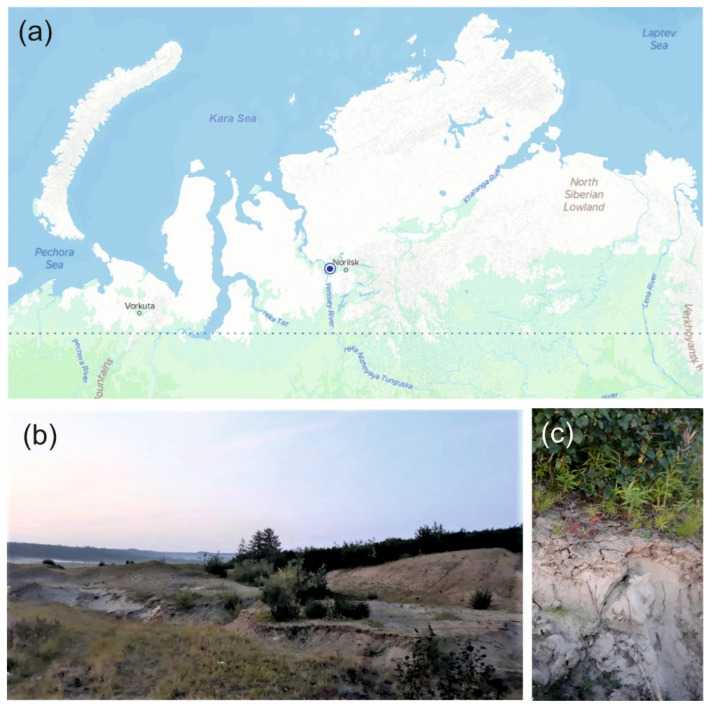
Location and some characteristics of the study site: (**a**)—location of the study site on the map of northern Russia; (**b**)—vegetation cover at the sampling site; (**c**)—soil profile.

**Figure 2 microorganisms-11-02800-f002:**
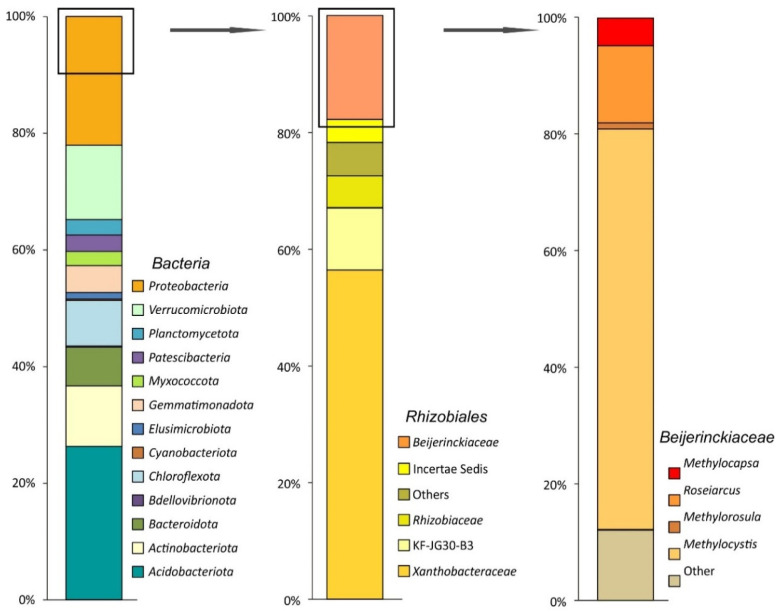
Bacteria community composition in a subarctic soil according to the results of Illumina-based sequencing of 16S rRNA genes. The composition is displayed at the following taxonomic levels (from left to right): the phylum level, the alphaproteobacterial order *Rhizobiales*, and the family *Beijerinckiaceae*. The relative abundance values represent averages of three replicate data sets.

**Figure 3 microorganisms-11-02800-f003:**
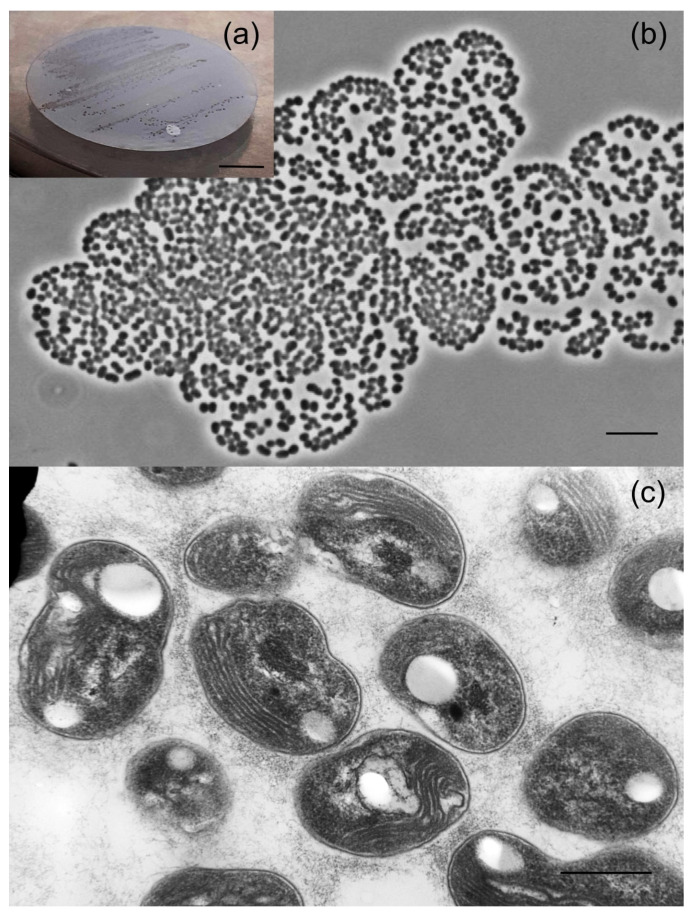
(**a**) Colony formation on the surface of polycarbonate filters after 2 months of incubation with ~30% (*v*/*v*) methane. Marker, 5 mm. (**b**) Cell morphology of strain D3K7. Marker, 5 μm; (**c**) Electron micrograph of an ultrathin section of methane-grown cells of strain D3K7. Marker, 200 nm.

**Figure 4 microorganisms-11-02800-f004:**
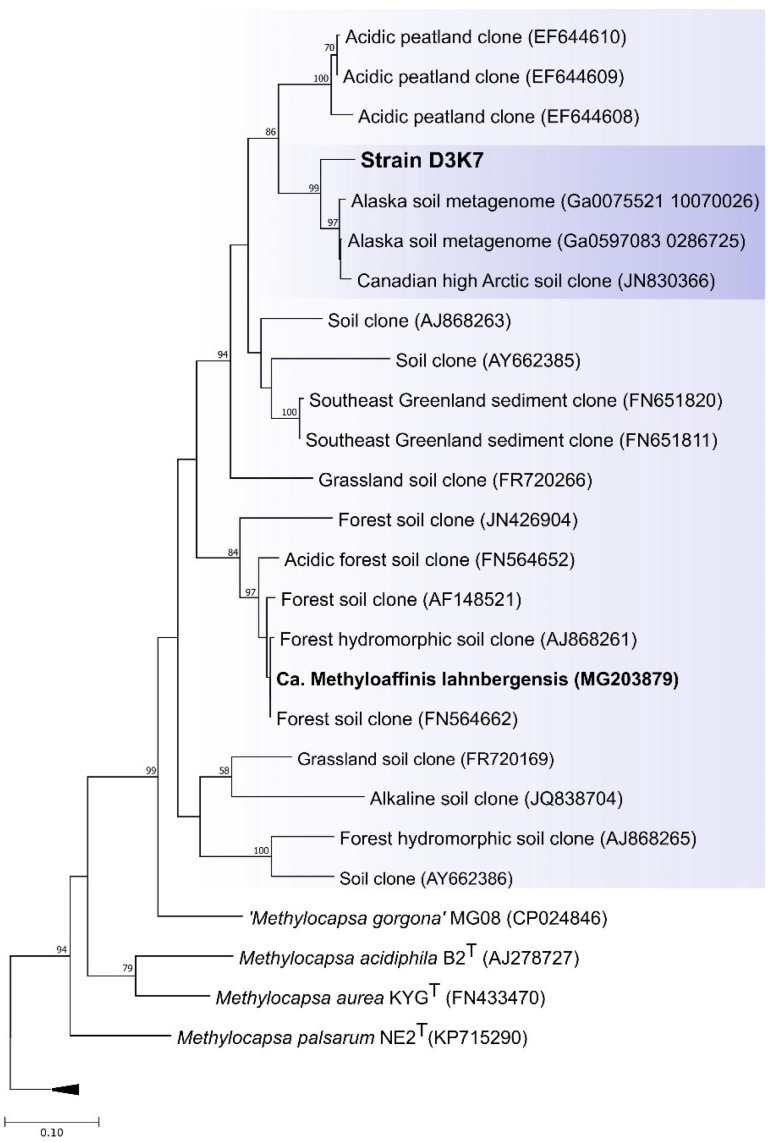
Phylogenetic dendrogram constructed based on the results of the comparative analysis of *pmoA* gene sequences from strain D3K7, other described species of the genus *Methylocapsa,* and uncultivated representatives of the USCα group. The evolutionary history was inferred by using the Maximum Likelihood method. A total of 449 positions were included in the final dataset. Evolutionary analyses were conducted in MEGA X [44]. Bootstrap values of > 50% are shown. The root (not shown) was composed of *pmoA* gene sequences from five gammapteobacterial methanotrophs: *Methylohalobius crimeensis* 10Ki^T^ (AJ581836), *Methylomonas methanica* S1^T^ (U31653), *Methylomonas paludis* MG30^T^ (HE801217), *Methylosoma difficile* LC2^T^ (DQ119047), and *Methylovulum miyakonense* HT12^T^ (AB501285). Bar, 0.1 substitutions per nucleotide position.

**Figure 5 microorganisms-11-02800-f005:**
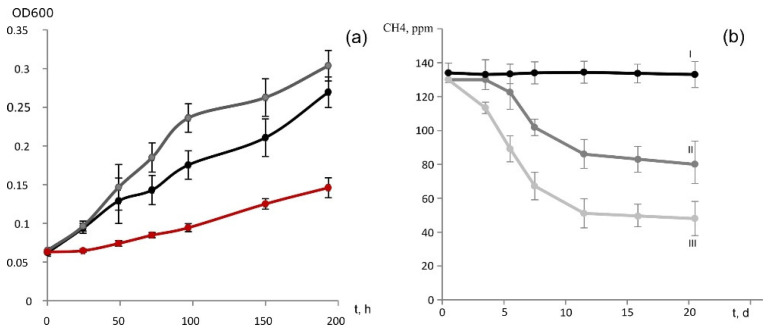
(**a**) Growth dynamics of strain D3K7 on methane (black), acetate (red) or in the presence of both methane and acetate (grey). (**b**) Dynamics of methane concentration in experimental flasks with liquid cultures of strain D3K7 supplied with methane (II) or both methane and acetate (III). Control vials (I) contain sterile liquid mineral medium. All data are means of triplicates.

**Figure 6 microorganisms-11-02800-f006:**
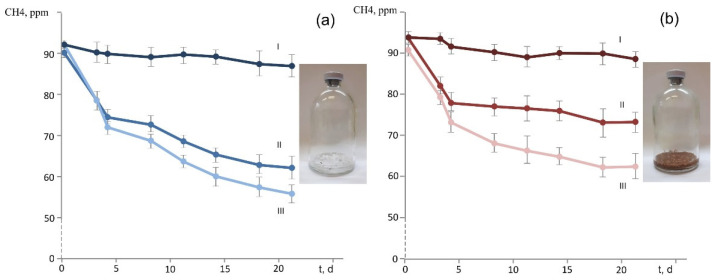
Dynamics of methane concentration in experimental bottles with adsorbent materials perlite (**a**) or vermiculite (**b**), which were used to simulate mineral particles in a well-aerated soil. Incubations were supplied with methane (II) or both methane and acetate (III). Control bottles (I) were supplied with methane but were not inoculated with cells of strain D3K7. All data are means of triplicates.

**Figure 7 microorganisms-11-02800-f007:**
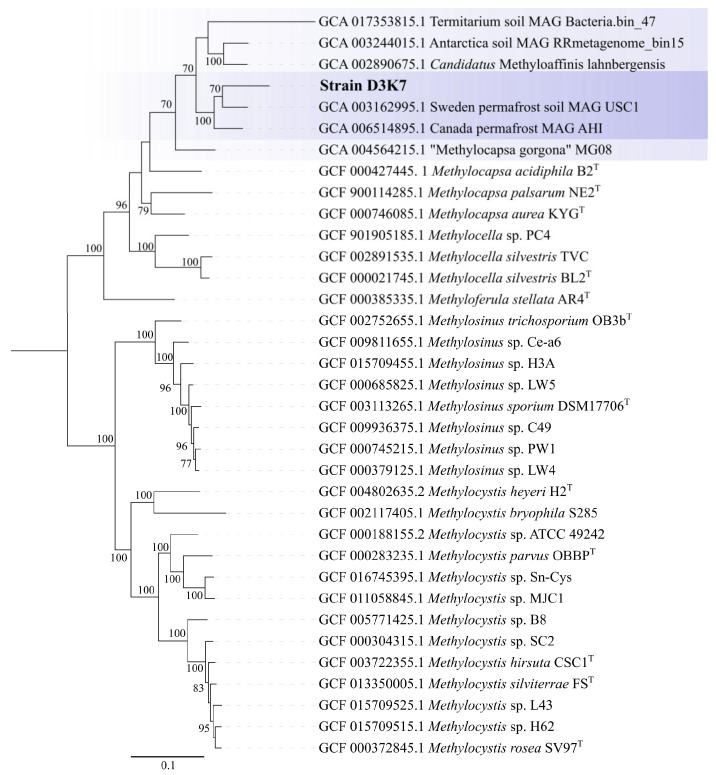
Genome-based phylogeny of strain D3K7. The clade of phylogenetically related USCα and USCα-like methanotrophs. The tree was constructed using the Genome Taxonomy Database Toolkit v2.0.0 [43]. The significance levels of interior branch points obtained in maximum-likelihood analysis were determined by bootstrap analysis (100 data re-samplings). Bootstrap values of >70% are shown. The root is composed of the family *Methylocystaceae* genomes (type II methanotrophs). Bar, 0.1 substitutions per amino acid position.

**Figure 8 microorganisms-11-02800-f008:**
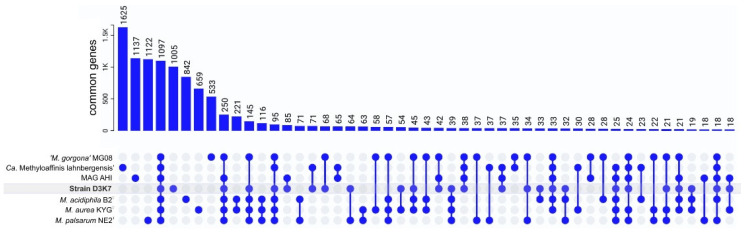
Distribution diagram of orthologous genes. Groups > 15 genes are shown. The level of amino acid sequence similarity >50% was used as a criterion for gene orthology.

**Figure 9 microorganisms-11-02800-f009:**
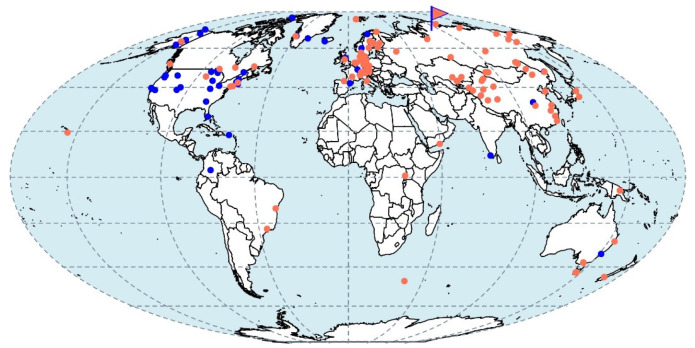
Global distribution of the D3K7-related lineage. The map indicates sampling locations of IMG and GenBank databases hits for *pmoA* genes with >85% identity (blue dots) and GenBank SRA database hits for 16S rRNA genes with >98% identity (coral dots). Strain D3K7 sampling location is indicated with a flag.

## Data Availability

Data is contained within the article and are deposited in Sequence Read Archive.

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
