# Peer review of "One Step Closer to Enigmatic USCα Methanotrophs: Isolation of a Methylocapsa-like Bacterium from a Subarctic Soil"

_microorganisms, 2023, doi:10.3390/microorganisms11112800_

Round 1
Reviewer 1 Report
Comments and Suggestions for Authors
This is an exciting study describing the second-only successful cultivation attempt of a bacterium capable of oxidising methane at atmospheric levels (or at least <100 ppmv). While, as the authors state, this new strain does not lie within the phylogenetic cluster of USC-alpha, it is so far the closest cultivar to the MAG of Candidatus Methyloaffinis lahnbergensis. Although the strain is quite similar to the relatively newly cultivated Methylocapsa gorgona MG08, it is idiosyncratic enough to merit its own focus. For example, the genome of the current strain, D3K7, has a significantly larger genome with many unique genes and genes shared with other methanotrophs but not with MG08. The experiments were carried out decisively, and the manuscript is well-written. I only have a few minor comments for the work, and I would highly recommend it for publication.
Roey Angel
Major comment:
Much of the part in the introduction about USC-alpha reads outdated in light of the relatively recent isolation of Methylocapsa gorgona MG08. Although the authors claim that its pmoA gene "does not correspond to the major group of PmoA sequences" (sic), this does not seem to be the case in Tveit et al. I think that this section just needs a bit of rewriting to not read like an unfolding historical account.
Minor comments:
Figure 1 is trimmed, and the bottom third is missing
L. 55. Here and elsewhere, PmoA should be pmoA and italicised
L. 100. Should be "Qubit"
L. 101. I guess you meant to write "molar" :)
L. 103. Should be "were used"
L. 104 Should be "were clustered by applying the VSEARCH plugin"
L. 105. The database is actually called SILVA 138 SSU
L. 136. Here and elsewhere, missing space between 37 and C
L. 137. Remove the space between the number and the % sign
Author Response
We would like to thank the referee for the positive opinion on our study and for the useful comments-suggestions. All of them were taken into account while preparing the revised manuscript version.
Сomment: Much of the part in the introduction about USC-alpha reads outdated in light of the relatively recent isolation of Methylocapsa gorgona MG08. Although the authors claim that its pmoA gene "does not correspond to the major group of PmoA sequences" (sic), this does not seem to be the case in Tveit et al. I think that this section just needs a bit of rewriting to not read like an unfolding historical account.
Response: Sorry, we disagree with this opinion. We do believe that the USCα story is not finished yet (otherwise what would be the reason for us to keep hunting for these elusive methanotrophs?). The importance of M. gorgona isolation is highlighted in the introduction. However, M. gorgona does not fall within the clade of “true” or originally discovered USCα methanotrophs. The latter is represented by Ca. Methyloaffinis lahnbergensis and some of the first environmental sequences retrieved from forest soils, such as clone RA14 (acc. AF148521). We have re-constructed our pmoA-based tree in order to include more representative environmental sequences in it (see revised Figure 4). It shows the same thing, namely, that M. gorgona clusters a bit separately from the originally discovered USCα methanotrophs. This situation is also true for our isolate, strain D3K7, although it is getting a bit closer to “true” USCαs (both on pmoA- and genome-based trees). The story will be finished when someone obtains and describes a pure culture of Ca. Methyloaffinis lahnbergensis or one of the closely related organisms. We have included the revised pmoA-based tree in the manuscript.
Сomment: Figure 1 is trimmed, and the bottom third is missing
Response: We apologize. The figure size was a bit too large. It has been optimized. Should be no problem now.
Сomment: L. 55. Here and elsewhere, PmoA should be pmoA and italicized
Response: We have corrected it everywhere in the text except of the paragraph in lines 249-253, where we compare translated amino acid sequences of PmoA from strain D3K3 to those of other methanotrophs.
Сomment: L. 100. Should be "Qubit"
Response: corrected.
Сomment: L. 101. I guess you meant to write "molar" :)
Response: you’re right. Done.
Сomment: L. 103. Should be "were used"
Response: corrected.
Сomment: L. 104 Should be "were clustered by applying the VSEARCH plugin"
Response: corrected.
Сomment: L. 105. The database is actually called SILVA 138 SSU
Response: corrected.
Сomment: L. 136. Here and elsewhere, missing space between 37 and C
Response: Corrected throughout the manuscript.
Сomment: L. 137. Remove the space between the number and the % sign
Response: Done.
Reviewer 2 Report
Comments and Suggestions for Authors
This is indeed one step closer to an isolate of upland soil cluster alpha, which is an exciting development in the field of environmental microbiology. The manuscript is valuable as it provides both new insights into the physiology of these enigmatic bacteria and also some evidence on how they can be cultivated using a static system. It’s very interesting that acetate enhanced methane oxidation, but it’s also curious that it didn’t seem to oxidise methane below 50 ppm (Fig 5b), but I don’t expect the authors to be able to answer this yet. Overall this appeared to be carefully performed and analysed, and it is clearly written. I mostly have minor comments.
I’m not sure if the results as presented support the statement in the abstract that D3K7-like methanotrophs colonize mostly northern terrestrial ecosystems. Figure 9 and the section on global distribution doesn’t show this, as you also state on L382-383. To my knowledge, none of the sites in reference 21 were boreal (L50-51). Also, Candidatus Methyloaffinis lahnbergensis was obtained from a temperate soil not a boreal soil (L57).
Minor comments
Change Sweeden to Sweden in the manuscript
Check that USCα has the greek alpha rather than roman ‘a’ when appropriate
L100, change Qubitn to Qubit?
L101, change moral to molar
L112, Perhaps you mean 8000 rpm not 8 rpm?
L224, refers to 2 months incubation, but Fig 3 legend says 14 days. Clarify if these were different experiments.
Fig 7, the branch below strain D3k7 is titled permaforst instead of permafrost.
Author Response
We would like to thank the referee for the positive opinion on our study and for the useful comments-suggestions. All of them were taken into account while preparing the revised manuscript version.
Сomment: I’m not sure if the results as presented support the statement in the abstract that D3K7-like methanotrophs colonize mostly northern terrestrial ecosystems. Figure 9 and the section on global distribution doesn’t show this, as you also state on L382-383. To my knowledge, none of the sites in reference 21 were boreal (L50-51). Also, Candidatus Methyloaffinis lahnbergensis was obtained from a temperate soil not a boreal soil (L57).
Response: This is a fair comment. We have revised the corresponding sentence in a following way: As shown by the global distribution analysis, D3K7-like methanotrophs are not restricted to polar habitats but inhabit peatlands and soils of various climatic zones.
Сomment: Change Sweeden to Sweden in the manuscript
Response: Done.
Сomment: Check that USCα has the greek alpha rather than roman ‘a’ when appropriate
Response: we have re-checked all symbols for correctness.
Сomment: L100, change Qubitn to Qubit?
Response: corrected.
Сomment: L101, change moral to molar
Response: done.
Сomment: L112, Perhaps you mean 8000 rpm not 8 rpm?
Response: You’re right. Corrected.
Сomment: L224, refers to 2 months incubation, but Fig 3 legend says 14 days. Clarify if these were different experiments.
Response: Thank you for noticing this discrepancy. The figures (a) and (b) refer to the experiment with filters. We now explain this in the figure caption. However, the cell morphology of strain D3K7 was the same after 14 days or 2 months of incubation.
Сomment: Fig 7, the branch below strain D3k7 is titled permaforst instead of permafrost.
Response: Thank you for noticing this mistake. It has been corrected.